DISCOVERY REPORT

# Evidence for genetically-based sperm discrimination in the vaginal tract of a primate species

Rachel M. Petersen[1¤a*], Lee (Emily) M. Nonnamaker[2¤b], Jaclyn A. Anderson[3], Christina M. Bergey[4], Christian Roos[5,6], Amanda D. Melin[3,7,8], James P. Higham[1]

1 Department of Anthropology, New York University, New York, New York, United States of America, 2 Department of Biological Sciences, University of Notre Dame, Notre Dame, Indiana, United States of America, 3 Department of Anthropology and Archaeology, University of Calgary, Calgary, Alberta, Canada, 4 Department of Genetics, Human Genetics Institute, Rutgers University, Piscataway, New Jersey, United States of America, 5 Primate Genetics Laboratory, German Primate Center, Leibniz Institute for Primate Research, Göttingen, Germany, 6 Gene Bank of Primates, German Primate Center, Leibniz Institute for Primate Research, Göttingen, Germany, 7 Department of Medical Genetics, University of Calgary, Calgary, Alberta, Canada, 8 Alberta Children's Hospital Research Institute, University of Calgary, Calgary, Alberta, Canada

¤a Current address: Department of Biological Sciences, Vanderbilt University, Nashville, Tennessee, United States of America
¤b Current address: Department of Biology, University of Florida, Gainesville, Florida, United States of America
* Rachel.m.petersen@vanderbilt.edu

## Abstract

Females influence offspring paternity through diverse pre- and post-copulatory mechanisms. Sperm discrimination—the differential physiological response to ejaculates based on male or sperm characteristics—can bias fertilization outcomes, but in vivo evidence of this process in large-bodied mammals is lacking. Here, in a study of nine females and four males, we tested whether two aspects of female physiology that affect sperm survival—vaginal immune response and pH—are modulated by male genetic makeup in a non-human primate, the olive baboon (*Papio anubis*). Our findings suggest post-copulatory differences in vaginal gene expression and pH, with the strongest immune responses and largest pH decreases, harmful to sperm, exhibited by females mating with genetically similar males. These findings are consistent with genetically-based post-copulatory mate discrimination, offering new insights into how interactions between male gametes and the female reproductive tract may shape conception probability in primates.

## Introduction

Characterizing the mechanisms and outcomes of sexual selection, and specifically mate choice, has been a major goal of evolutionary biologists [1–4]. Female mate choice can occur both prior to copulation in the form of behavioral mating biases, or

**Data availability statement:** The sequencing reads generated in this study have been submitted to the NCBI Sequence Read Archive (SRA; https://www.ncbi.nlm.nih.gov/sra) under accession numbers PRJNA875430 (ddRAD sequences) and PRJNA1232174 (RNA sequences), and in GenBank (https://www.ncbi.nlm.nih.gov/genbank/) under accession numbers OP375715-OP375798 (MHC sequences). Counts matrices, pH measurements, and metadata needed to rerun all code are available at Zenodo (https://zenodo.org/records/14976902). Code is available at Zenodo (https://zenodo.org/records/18705035) and at RMP's personal GitHub page: www.github.com/rachpetersen/cryptic_choice_anubis.git. Data used to generate figures are available in S1–S5 Data.

**Funding:** This study was funded by: National Science Foundation Doctoral Dissertation Research Improvement Grant (grant no. 1826804) to RMP and JPH, Wenner-Gren Foundation Dissertation Fieldwork Grant (grant #9921) to RMP, Leakey Foundation Research Grant to RMP, Animal Behavior Society Student Research Grant to RMP, Primate Society of Great Britian Primate Research Grant to RMP, International Primatological Society Research Grant to RMP, American Society of Mammologists Grant-in-Aid to RMP, Society for Integrative and Comparative Biology Research Grant to RMP, Sigma Xi Grant-in-Aid of Research to RMP, New York University Intramural Funds to JPH, and the Canada Research Chairs program (grant no. 950-231257) to ADM. The funders had no role in study design, data collection and analysis, decision to publish, or preparation of the manuscript.

**Competing interests:** The authors have declared that no competing interests exist.

**Abbreviations:** BEB, Bayes Empirical Bayes; BLAST, basic local alignment search tool; CFC, cryptic female choice; ddRAD-seq, double digest restriction-site associated DNA sequencing; DE, differentially expressed; EI, eosinophilic index; GSEA, gene set enrichment analysis;

after copulation in the form of fertilization biases, a process termed cryptic female choice (CFC) [5–8]. To date, empirical evidence demonstrating in vivo CFC in mammals is concentrated in rodent taxa [9,10]. However, the heightened maternal investment and prolonged offspring care common to large-bodied mammals, as well as discrepancies between mating observations and genetic paternity, suggest that CFC may be widespread [11–13]. Nonetheless, investigating these processes in species which share aspects of their reproductive physiology with humans, such as other primates, is likely critical for improving our understanding of human infertility.

Studies indicate that the female reproductive tract can discriminate between sperm cells based on their genetic material [14–16], providing a potential mechanism for genetically-based CFC. In mammals, in vitro experiments in mice show higher fertilization success for sperm from more distantly related males [17], and artificial insemination experiments in pigs reveal dramatic shifts in oviductal gene expression in response to sex-sorted X- versus Y-chromosome-bearing sperm [18]. In humans, in vitro experiments have shown both differential sperm responsiveness to follicular fluid and differential gene expression in vaginal epithelial cells in response to seminal fluid, however, how these responses relate to the genetic make-up of the egg and sperm remains uncertain [19,20]. The major histocompatibility complex (MHC) is a highly polymorphic genomic region involved in pathogen identification and immune response regulation. It is also an attractive candidate target of CFC due to its prior implicated role in mate choice and important contribution to reproductive success [21–23]. While pre-copulatory MHC-based mate preferences are well documented across taxa, including non-human primates [24–29], the role of the MHC in post-copulatory sexual selection remains largely unexplored. Evidence for MHC-driven sperm selection is limited to a handful of studies in rodents, fish, and birds [30–33], with no documented evidence in primates, despite its potential relevance to human fertility.

In this study, we aimed to explore potential mechanisms of CFC in a non-human primate, the olive baboon (*Papio anubis*). Olive baboon females mate with multiple males across their ovarian cycle, however, males often attempt to monopolize access to fertile females through mate guarding. These consortships, in which a male closely associates with and guards a female, can persist for several hours to multiple days, during which time the ejaculate from only a single male may be present in the female's reproductive tract [34]. Furthermore, females energetically invest greatly in each offspring, and experience a relatively slow reproductive rate, providing conditions that are likely to promote selection for CFC [35]. We focused on vaginal pH and gene expression, as these may contribute to sperm survival [36,37] and can be characterized following mating in unanesthetized individuals using positive reinforcement training. We conducted both genome-wide reduced representation DNA sequencing and MHC genotyping on four intact males and nine parous females and strategically paired each male with 2–3 females to encompass a broad range of genetic diversity and complementarity (i.e., similarity) values across mating dyads. We first characterized vaginal pH and gene expression across the cycle in the absence of mating and used these samples as baseline comparisons for post-copulatory responses. We asked how vaginal gene expression and pH changes: (1) across female ovarian

LFSR, local false sign rate; MHC, major histocompatibility complex; PAML, Phylogenetic Analysis by Maximum Likelihood; PSS, positively selected sites; RAxML, randomized axelerated maximum likelihood; stMLH, standardized multi-locus heterozygosity; VIF, variance inflation factor; WBCs, white blood cells.

cycle phases; (2) in response to mating; and (3) in relation to the genetic diversity and complementarity of the mating male. We hypothesized that females will exhibit a stronger immune response and lower vaginal pH, both potentially harmful to sperm survival, after mating with males who are less genetically diverse and complementary. We predicted this pattern based on the selective pressures favoring offspring with greater genetic diversity, particularly at the MHC, while reducing the risks associated with inbreeding.

## Results

### Vaginal gene expression varies across the ovarian cycle

We determined the timing of ovulation based on vaginal cytology (Fig 1A; see Materials and methods). We identified a 5-day fertile phase, a 5-day pre-fertile phase, a 5-day post-fertile phase, and classified the remainder of the cycle as the non-fertile phase [38–41]. We analyzed 32 non-copulatory vaginal RNA samples from eight females (8 per cycle phase) and 275 non-copulatory pH measurements from nine females ($68.8 \pm 21.8$ s.d. per cycle phase; Fig 1A; additional details on dataset composition provided in Table A in S2 Appendix).

To understand baseline vaginal gene expression, we performed differential gene expression analyses with robust empirical Bayes moderation followed by adaptive shrinkage [42,43]. We included cycle phase as the predictor variable, female ID as a blocking factor (i.e., a random effect), and RNA quality (RIN) as a covariate. As some samples were taken prior to male introduction into the enclosure, we also included "male presence" as a covariate (a binary yes or no variable). We found 2,480 differentially expressed (DE) genes between the non-fertile versus pre-fertile phase (2,330 at LFSR < 0.05, 2,050 at LFSR < 0.01), 2,537 between the non-fertile versus fertile phase (2,403 at LFSR < 0.05, 2,174 at LFSR < 0.01), and 2,277 between the non-fertile versus post-fertile phase (2,080 at LFSR < 0.05, 1,675 at LFSR < 0.01; Fig 1B), with many of these shared across the pre-fertile, fertile and post-fertile phases ($n = 1,974$; Fig 1C and Table B in S2 Appendix). We performed gene set enrichment analysis (GSEA) to describe the biological functions of DE genes and found that the most strongly upregulated pathways in the fertile phase involve G protein-coupled receptor activity and ion transmembrane transport, and the most strongly downregulated pathways include positive regulation of the inflammatory response, phagocytic vesicles, and cell adhesion (Fig 1D and Fig A in S1 Appendix and Table C in S2 Appendix). For example, *TLR2,* a gene involved in the positive regulation of the inflammatory response, was suppressed in the fertile phase (Fig 1E) and *SLC4A8*, a gene involved in ion transmembrane transport, was activated in the fertile phase (Fig 1F).

To assess changes in vaginal pH across the cycle, we used robust linear mixed models [44] including female ID as a random effect and average temperature, which is known to impact pH readings, as a covariate [45,46]. We did not find statistically significant differences in vaginal pH across cycle phases (Fig B in S1 Appendix and Table D in S2 Appendix).

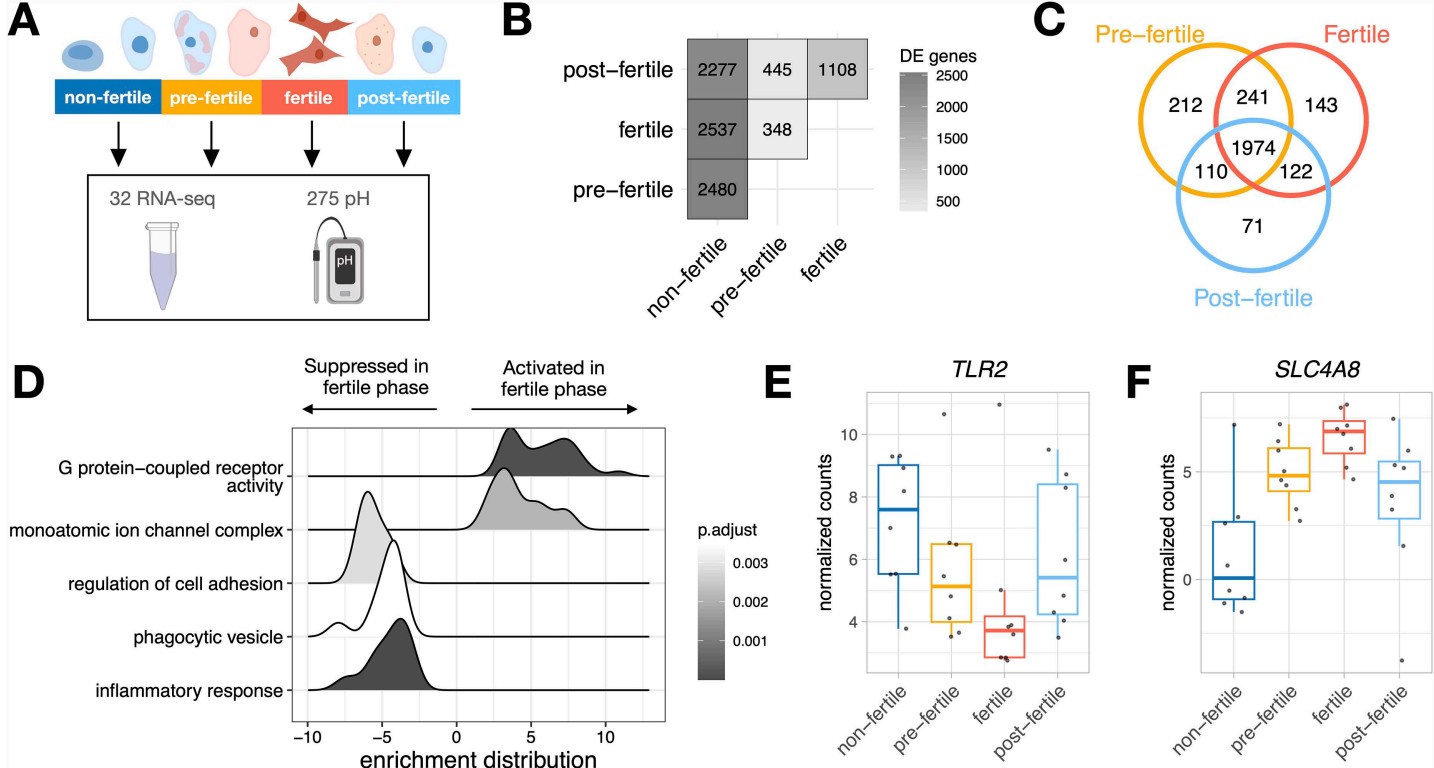

**Fig 1. Differential gene expression measured across ovarian cycle phases. (A)** We analyzed 32 RNA-seq samples and 275 pH measurements taken across the four cycle phases as determined by vaginal cytology; **(B)** The number of differentially expressed (DE) genes across each phase comparison. The largest differences in gene expression were observed when comparing the non-fertile phase to the pre-fertile, fertile, and post-fertile phases; **(C)** Numerous DE genes were unique to particular cycle phases (i.e., 143 DE genes were unique to the fertile phase), while others were shared across two or more phases (i.e., 1,974 DE genes were shared across the pre-fertile, fertile, and post-fertile phases); **(D)** Enrichment distributions showing the ranked distribution of genes in the top 5 over/underrepresented gene sets in the fertile phase; **(E)** Normalized read counts of *TLR2*, a gene involved in the positive regulation of the inflammatory response and suppressed in the fertile phase; **(F)** Normalized read counts of *SLC4A8,* a gene involved in ion transmembrane transport and activated in the fertile phase. The data underlying this figure are provided in S1 Data. Artwork by LMN.

## Vaginal gene expression and pH indicate responses to copulation

We analyzed 25 post-copulatory RNA samples from six females and 15 post-copulatory pH measurements from five females (described in Table A in S2 Appendix), both collected four hours after an observed copulation with ejaculation to maximize potential changes in gene expression [19]. All post-copulatory samples were collected during the pre-fertile, fertile, or post-fertile phases, and compared to non-copulatory samples taken from the same females during those same three phases when copulation had not been observed that day and there was no evidence of a sperm plug (non-copulatory RNA samples: $n_{non-cop} = 30$, pH measurements: $n_{non-cop} = 47$; Fig 2A). To account for male-derived RNA present in the vagina, we performed RNA-seq on two semen samples from each male collected opportunistically following masturbation (N= 8 samples total). From these samples, we identified 1,442 genes highly expressed in semen (average expression of >20 cpm, Table E in S2 Appendix), and removed these genes from all subsequent post-copulatory gene expression analyses.

We performed differential expression analyses using robust estimation followed by adaptive shrinkage, with post-copulatory status (yes or no) as the predictor variable, dyad ID as a blocking factor, and RIN, cycle phase, and male presence as covariates. We identified 941 DE genes in post-copulatory versus non-copulatory samples (715 at LFSR < 0.05,

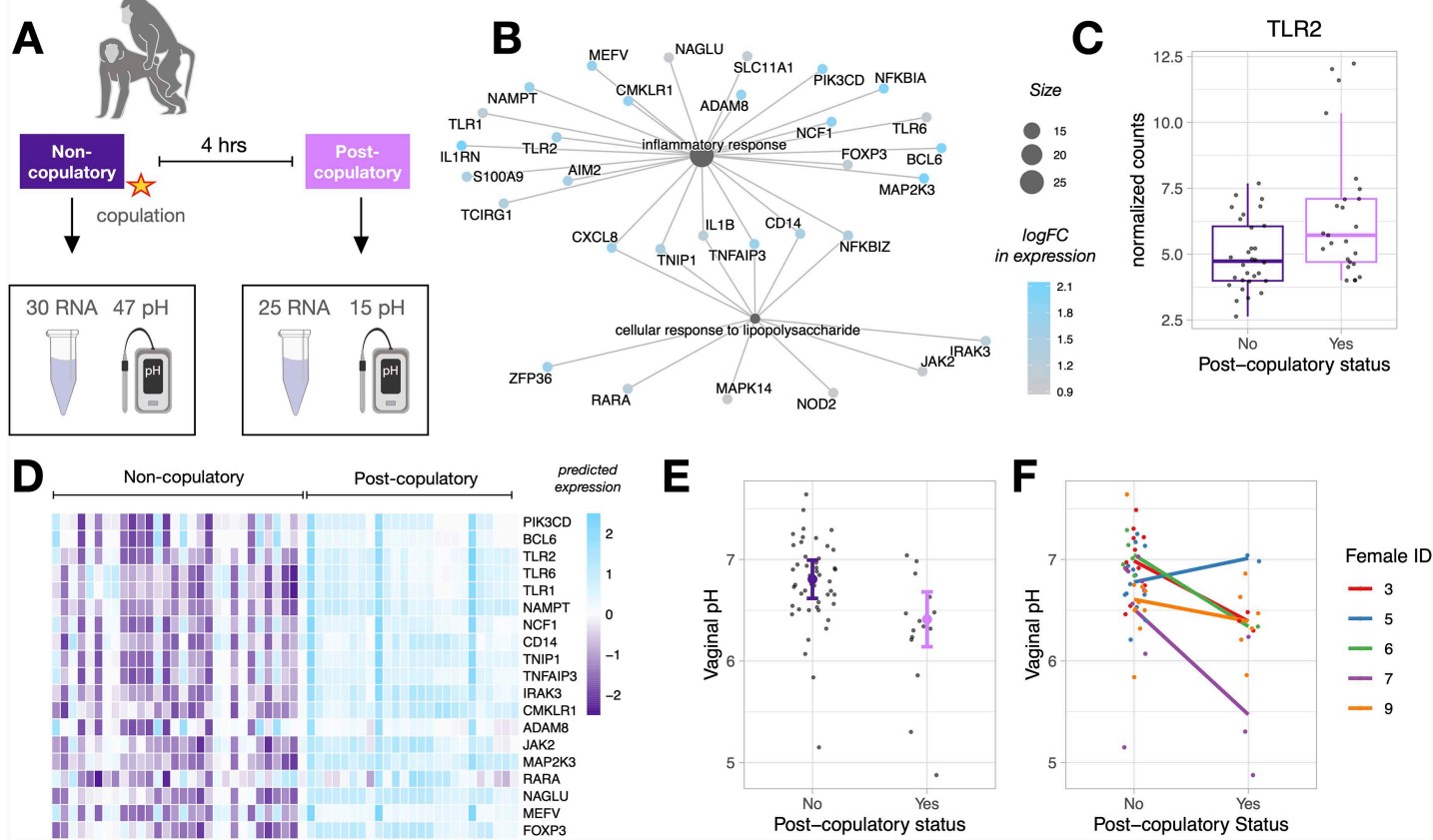

**Fig 2. Differential gene expression and pH in response to copulation. (A)** We analyzed 30 RNA-seq samples and 47 pH measurements taken when there was no evidence of recent copulation, and 25 RNA-seq samples and 15 pH measurements taken 4 hours following copulation; **(B)** Genes within two immune-related GO pathways upregulated in post-copulatory samples. Gene set nodes are sized based on the number of genes within them, genes nodes are colored by their log fold change (logFC) in expression; **(C)** Normalized counts of *TLR2* in non-copulatory vs. post-copulatory samples; **(D)** Predicted expression of immune system related genes that are differentially expressed in post-copulatory vs. non-copulatory samples. Columns represent samples (left = non-copulatory, right = post-copulatory), rows represent genes, and cell color represents the predicted increase (blue) vs. decrease (purple) in expression, scaled across each row; **(E)** Vaginal pH was significantly lower in post-copulatory compared to non-copulatory samples. Colored points and error bars represent model predictions ± one standard error and black points represent raw data; **(F)** Females show non-uniform patterns in the direction and magnitude of pH change between non-copulatory and post-copulatory samples. The data underlying this figure are provided in S2 Data. Artwork by LMN.

383 at LFSR < 0.01; Table F in S2 Appendix). DE genes were enriched for two ontology pathways, both of which are involved in immune system processes (Fig 2B and Table G in S2 Appendix). These enriched pathways include well-described genes that regulate chemokine signaling, such as *TLR2* (Fig 2C), and are generally predicted to have higher expression in post-copulatory versus non-copulatory contexts (Fig 2D).

To assess alterations in vaginal pH following copulation, we used robust linear mixed models including dyad ID as a random effect and cycle phase and average temperature as covariates. Although we did not find an association between cycle phase and pH in our dataset, we included cycle phase as a covariate due to previous work observing lower vaginal pHs around the time of ovulation in baboons and humans [47,48]. We found that post-copulatory pH measurements were significantly lower than non-copulatory measurements (estimate = −0.39, SE = 0.12, *p* = 0.001; Table H in S2 Appendix and Fig 2E), with substantial individual variation in the magnitude of pH change following copulation (Fig 2F). Although variance tests can be sensitive to small sample sizes, we nonetheless detect significantly greater variance in pH among

post-copulatory samples compared to non-copulatory ones (Breusch-Pagan test: $p = 0.05$), which aligns with our initial hypothesis and supports the use of an interaction model to test for the role of male genetic diversity and complementarity in moderating post-copulatory vaginal pH.

**Male genetic diversity and complementarity modulate post-copulatory vaginal gene expression and pH**

To explore whether post-copulatory vaginal gene expression and pH are modulated by male genetic makeup, we estimated both genome-wide and MHC diversity and complementarity. We used double digest restriction-site associated DNA sequencing (ddRAD-seq) to estimate standardized multi-locus heterozygosity (stMLH) and kinship to approximate genome-wide diversity and complementarity, respectively. We used amplicon sequencing of the antigen-binding cleft of four MHC loci (2 class I: A and B, and 2 class II: DQA and DRB) to calculate MHC diversity as the number of class I and class II alleles and complementarity as the proportion of shared alleles between dyads. We paired males and females based on their relative genetic compatibility to produce mating dyads with kinships ranging from −0.18 to 0.24 and MHC complementarity ranging from 10% to 40% (class I loci) and 0% to 40% (class II loci). We also characterized biologically relevant MHC "supertypes" based on amino acid polarity at positively selected sites within the antigen-binding cleft (detailed methods in [49]) to determine supertype-based diversity and complementarity. In total, we tested five measures of male diversity and five measures of complementarity between each mating dyad, summarized in Table I in S2 Appendix.

We performed 10 separate differential gene expression analyses, one for each measure of male genetic diversity or complementarity, applying robust estimation and adaptive shrinkage. Each model was constructed with dyad ID as a blocking factor, RIN, cycle phase, and male presence as covariates, and an interactive effect between post-copulatory status (yes or no) and male genotype as the predictor variable. Measures of male MHC diversity and complementarity were associated with an excess of low p-values relative to the null expectation, suggestive that male MHC diversity and complementarity broadly influence gene expression (Fig 3A and 3B). Complementarilty as measured using alleles was associated with stronger deviations from the null expectation compared to supertypes (Fig 3B). Genome-wide diversity (stMLH), in contrast, was not as strongly associated with gene expression changes (Fig 3A). Postcopulatory expression of 456 genes was associated with male MHC allele or supertype diversity, meeting both an LFSR < 0.1 and family-wise error rate (FWER)-adjusted $p < 0.05$ criteria (Table J in S2 Appendix). Although the different MHC metrics did not share any of the same significant genes, GSEA revealed that male MHC I allelic diversity and MHC II supertype diversity were both positively associated with the expression of genes involved in RNA polymerase activity and intercellular signaling pathways (Table L in S2 Appendix). Likewise, we identified 590 genes whose post-copulatory expression was associated with either MHC or genome-wide complementarity at an LFSR < 0.1 and FWER-adjusted $p < 0.05$, representing 25 and 17 GSEA pathways, respectively (Tables K and L in S2 Appendix). Once again, each metric was associated with a unique set of significant genes, however, GSEA revealed that both MHC I and MHC II allelic complementarity were both positively associated with pathways involved in immune response and cellular signaling (Fig 3C and Figs C and D in S1 Appendix and Table L in S2 Appendix). For example, *MAP3K2*, which functions in the MAP kinase signaling pathway and has been implicated in the activation of NF-κB and downstream cytokine production, is expressed more in females who mated with males with whom they share a greater number of MHC I alleles (i.e., low complementarity) in comparison to females who mated with males with whom they share fewer MHC I alleles (i.e., high complementarity; Fig 3D). All genotype-dependent differential expression and GSEA results are summarized in Table M in S2 Appendix.

To evaluate the robustness of our findings, we conducted a leave-one-out sensitivity analysis in which we iteratively excluded a single mating dyad from our dataset and re-ran the differential gene expression analysis. For each iteration, we tested the interaction between post-copulatory status and MHC I or MHC II allelic complementarity—two genotype features that yielded the strongest initial associations. Across iterations, we consistently observed a substantial number of DE genes associated with MHC I allelic complementarity (range: 203−1,191, mean = 837.2, SD = 343.1, LFSR < 0.1) indicating that this association is not driven by any single dyad. Differential expression linked to MHC class II allelic complementarity

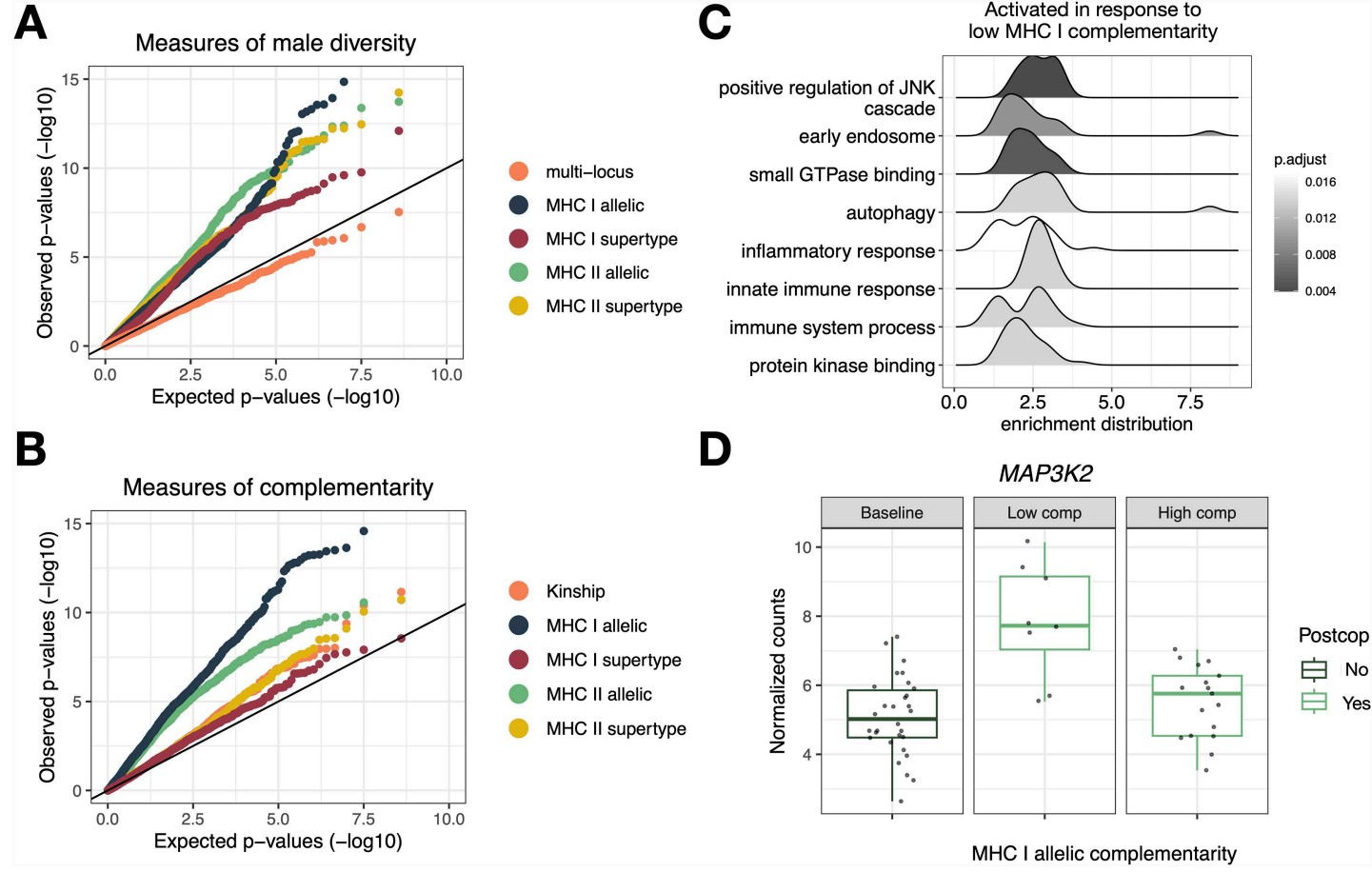

**Fig 3. Post-copulatory vaginal gene expression in relation to male diversity and complementarity in five mating dyads. (A)** and **(B)** Quantile–quantile (Q–Q) plots comparing observed to expected p-values for gene expression association with measures of male diversity (A) and complementarity (B). Low *p*-values are highly enriched in our observed data compared to the null expectation (black line on *x* = *y*) when assessing the effect of male MHC diversity (but not genome-wide diversity) and MHC I and II allelic complementarity; **(C)** Enrichment distributions showing the ranked distribution of genes in the top 8 overrepresented gene sets that are upregulated in expression with low MHC I allelic complementarity; **(D)** Expression of *MAP3K2*, a gene which plays a key role in phagocytosis, in a non-copulatory context (left panel) and in a post-copulatory context subset by the degree of MHC I allelic complementarity (high complementarity: sharing < 30% of alleles, low complementary: sharing > 30% of alleles). The data underlying this figure are provided in S3 Data.

was more variable, yet still consistently present across dyads (range: 86–1,006, mean = 456.3, SD = 317.3, LFSR < 0.1), indicating that our MHC class II results may be more sensitive to the individual dyads included in the analysis. Future studies with larger sample sizes will be necessary to validate these results.

To assess how male genetic diversity and complementarity modulates post-copulatory vaginal pH, we again fit 10 separate models, one for each measure of male genetic diversity or complementarity. We included dyad ID as a random effect, cycle phase, and average temperature as covariates, and an interaction between post-copulatory status and male genotype as the predictor variable. We found a significant interaction for three measures of genetic complementarity: kinship, class II allelic complementarity, and class II supertype complementarity (Table N in S2 Appendix). For all three metrics, the largest drops in post-copulatory vaginal pH are observed among females mating with genetically similar males (i.e., low complementarity), and the smallest drops (or potential increases) in vaginal pH are observed among females mating with genetically dissimilar males (i.e., high complementarity; Fig 4). Model fit was evaluated with Akaike's Information Criterion

PLOS Biology

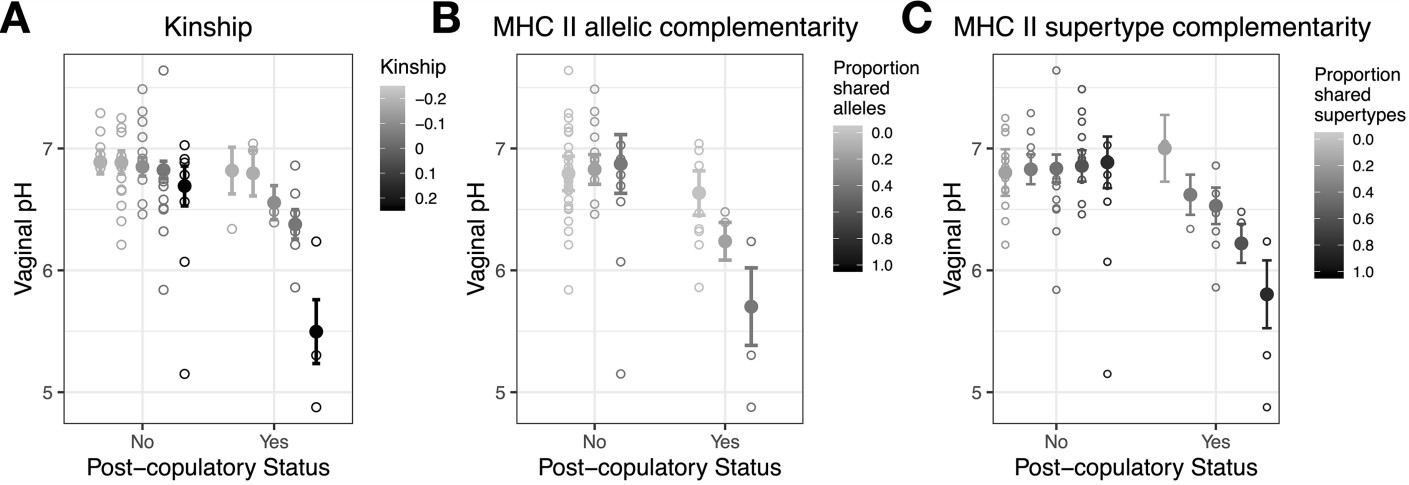

**Fig 4. Post-copulatory vaginal pH in relation to male genetic complementarity.** Model predictions illustrating the interaction between genetic complementarity and post-copulatory status in predicting post-copulatory vaginal pH, with the lowest post-copulatory pH observed among females mating with males with high degrees of kinship **(A)** and low degrees of MHC class II allelic **(B)** and supertype **(C)** complementarity. Filled points and error bars represent model predictions ± one standard error and open points represent raw data. The data underlying this figure are provided in S4 Data.

(AIC). For all three significant models, inclusion of the interaction term significantly improved model fit compared to simplified models that did not include male genotype (dAIC > 2). To assess whether our model estimates were driven by particular mating dyads, we refit models testing for the effect of kinship, MHC II allelic complementarity, and MHC II supertype complementarity using the leave-one-out method (see Materials and methods). Our results were generally recapitulated across iterations (Fig E in S1 Appendix). Although smaller sample sizes generate larger standard errors, all model estimates trended in the same direction as the model which included all mating dyads.

## Discussion

Together, our findings suggest that aspects of female reproductive physiology can respond differentially to male inseminations, providing preliminary support for genetically-based sperm discrimination—a potential mechanism by which post-copulatory mate choice may occur. Vaginal immune responses can protect females from infection, but these processes may need to be carefully regulated mid-cycle to accommodate exposure to paternal-derived molecules [50–52]. Our dataset supports this hypothesis, revealing a mid-cycle suppression of immune-related genes. In contrast, these same immune pathways show heightened expression post-copulation, with the magnitude of this response linked to male genetic characteristics. The striking convergence on similar pathways influenced by genotype at MHC class I and class II loci presents a particularly compelling case that vaginal responses may contribute to CFC, especially given the absence of a strong correlation between genetic diversity at these loci in this population [49]. The female immune system poses a potential detriment to sperm survival through processes enriched following mating with genetically similar males [53], however, a strong immune response may also prime the female reproductive tract for implantation [54–56], and future studies will be needed to distinguish between vaginal immune responses promoting and antagonizing successful conception [57–59]. Lastly, we find that post-copulatory vaginal pH is strongly associated with male genetic complementarity, with the largest drops—detrimental to sperm survival—occurring after mating with genetically similar males, suggesting that vaginal pH dynamics may also serve as a mechanism of CFC alongside changes in vaginal immune response.

This study furthers our understanding of how the mammalian vaginal environment, experienced as a first point of contact between male gametes and the female reproductive tract, may mechanistically contribute to sperm success and

potential offspring genotypes. While these findings are based on a limited dataset and should be interpreted as such, they provide intriguing support for a potential mechanism of non-directional sexual selection driven by genetic complementarity (i.e., non-additive mate choice). In this context, genotype-by-genotype interactions would drive CFC, dampening consistent directional shifts in allele frequencies over time across the population. Future work with larger sample sizes will be needed to confirm these patterns, as well as to explore male-driven sexually antagonistic strategies that circumvent female-mediated processes. As our close evolutionary relatives, we are excited by the potential of future non-human primate research to clarify the molecular underpinnings and evolutionary origins of variation in conception probability in humans as well as other mammals.

## Materials and methods

### Study subjects and experimental design

We worked with a population of captive olive baboons housed at le Centre National de La Recherche Scientifique Station de Primatologie (CNRS SdP), in Rousset, France. Study subjects consisted of 13 individuals, 4 intact males and 9 parous females. We created 4 small study groups composed of 1 male and either 2 or 3 females (3 groups contained 2 females, 1 group contained 3 females). Females were not on any form of contraception. Prior to the start of this study, each group of 2–3 females was housed with a vasectomized male and none of the females were pregnant. To create our study groups, CNRS SdP staff relocated the resident vasectomized male and allowed each group of females to live without a male for one month (the length of one ovarian cycle), during which time females underwent positive reinforcement clicker training to present their hindquarters for vaginal swabbing and pH measurement. After one month, the resident ethologist managed the introduction of an intact male by introducing males to females first from an adjacent enclosure, allowing visual and olfactory interaction for 2–3 days prior to physical introduction. After the intact male was physically introduced, we collected data on each group for two months (the duration of two ovarian cycles). During the course of the study, two females became pregnant, and all data collected from these females following the ovulation window in which they conceived was discarded. All manipulations and treatments received ethical approval from the Ministry of Higher Education, Research and Innovation in France (APAFIS#15021-2018051115066627) and the NYU University Animal Welfare Committee (18-1504), and were compliant with the European Science Foundation animal-handling guidelines to minimize pain and distress.

### Genome-wide and MHC genotyping

We utilized genotyping data generated as part of a previous study assessing the concordance between genome-wide and MHC diversity and complementarity in olive baboons living at CNRS SdP [49]. Detailed library preparation, sequencing, and bioinformatic methods are described in detail in [49] and are described in brief below.

We extracted DNA from whole blood using the Qiagen QIAamp DNA mini kit ($N=4$; $n_{female}=3$, $n_{male}=1$) or the GEN-IAL First-DNA All tissue kit ($N=9$; $n_{female}=6$, $n_{male}=3$) following manufacturer's instructions. To assess genome-wide diversity and complementarity between dyads, we performed double digest restriction-site associated DNA sequencing (ddRAD-seq). We prepared ddRAD-seq libraries following [60]. We digested 1 µg of DNA using restriction enzymes (SphI and MluCI), and size selected for 185 (±19) bp fragments using the Blue Pippin System. We ligated Illumina platform adapters, indexed samples using NEBNext Multiplex Oligos for Illumina sequencing, and sequenced on the Illumina HiSeq 2500 platform using one lane and 150 bp PE reads. We excluded low-quality reads and reads not containing both enzyme cut sites, mapped reads to the olive baboon reference genome (Panu v3) using the bwa mem aligner with default parameters [61], and performed shared SNP calling using the STACKS v2 reference pipeline [62]. We required that a locus be sequenced in at least 80% of individuals to be included in the final SNP set, and excluded SNPs in strong linkage disequilibrium ($r^2 > 0.5$) using PLINK [63]. Our final SNP set consisted of 35,509 SNPs to be used in the calculation of stMLH [64] for each individual, and for genome-wide complementarity between each dyad (i.e., kinship). We calculated stMLH by

dividing the proportion of genotyped loci at which an individual was heterozygous by the population mean heterozygosity at all genotyped loci, using the "Rhh" package in R [65]. We calculated kinship between each dyad using the relationship inference algorithm in the software package KING v2.2.4 [66].

To assess MHC diversity and complementary between dyads, we performed PCR amplification of the functionally important antigen-binding regions of two class I MHC loci (A and B) and two class II MHC loci (DQA and DRB). We chose to assess both class I and class II loci because they encode for molecules that are present on different cell types and perform unique functions: class I molecules are present on the surface of nearly all nucleated cells and bind to intracellular pathogens such as viruses, and class II molecules are found on the surface of antigen-presenting cells and bind to extracellular pathogens such as bacteria [67]. Moreover, previous results suggest that in this population, genetic diversity at MHC class I loci is not strongly associated with diversity at class II loci, meaning that cryptic choice mechanisms may favor diversity and/or complementarity at one locus and not the other [49]. We targeted a 195 bp segment within the α1 domain of the class I receptor types (MHC-A and -B), a 188 bp segment within the α1 domain of the DQA receptor, and a 252 bp segment within the β1 domain of the DRB receptor. These sequences make up part of the antigen-binding cleft of each receptor type, and amino acid variation within these regions can result in variable pathogen recognition and binding [68]. We amplified the desired sequences using the MilliporeSigma FastStart High Fidelity PCR System and primers described in Table O in S2 Appendix. Following amplification, we selected the amplicon of the appropriate length using gel electrophoresis and band excision, performed an indexing PCR using Hot Start Pfu DNA Polymerase, and sequenced on the Illumina MiSeq platform with v2 chemistry and 200 bp PE reads. Following sequencing, we trimmed and mapped sequences to MHC-A, -B, -DQA, and -DRB sequences taken from the IPD-MHC database and for each individual retained unique MHC sequences that had over 1,000 reads and were also present at >5% copy number in another individual. We calculated MHC diversity and complementarity for class I and class II loci separately. We calculated an individual's MHC allelic diversity as the number of unique MHC alleles and calculated MHC allelic complementarity as the number of MHC alleles shared between two individuals divided by the total number of unique MHC alleles possessed by the two individuals in total.

## Identification of MHC supertypes

To support the potential biological relevance of our measures of MHC diversity and complementarity, we additionally identified MHC supertypes based on the physiochemical properties of the amino acids involved in antigen binding and calculated MHC diversity and complementarity for each dyad at the supertype level. To do so, we followed methods from [69], which are described briefly below and in detail for this specific dataset in [49]. First, we identified positively selected sites (PSS) within the antigen-binding region of each MHC locus by comparing rates of synonymous (dS) to nonsynonymous (dN) nucleotide substitutions in protein-coding regions using methods described by [70]. To do so, we determined sequence reading frames by performing an alignment to published sequences in the IPD-MHC database using NCBI's basic local alignment search tool (BLAST). We translated aligned sequences in R using the package 'seqinr' [71] and performed multiple protein sequence alignment in MAFFT v.7 [72]. We converted protein alignments into codon alignments using PAL2NAL v.14 [73], and constructed a phylogenetic tree of the alignments using randomized axelerated maximum likelihood (RAxML) [74] and a generalized time reversible (GTR) GAMMA substitution model, with the best-scoring tree selected using 100 bootstrap iterations. We then computed substitution rate ratios (dN/dS) by inputting the PAL2NAL codon alignment and RAxML tree into the CODEML program within the Phylogenetic Analysis by Maximum Likelihood (PAML) package [75]. This software identifies statistically significant PSS using the Bayes Empirical Bayes (BEB) analysis computed under NSsite model 8 [76]. Next, we aligned the amino acids associated with each PSS and described the physiochemical properties of each site in the form of five z-descriptors: $z_1$ (hydorphobicity), $z_2$ (steric bulk), $z_3$ (polarity), $z_4$, and $z_5$ (electronic effects) [77]. We compiled a mathematical matrix containing the five z-scores of each PSS of each allele and performed an agglomerative hierarchical clustering analysis using Euclidian distance and the average linkage method

with the R function 'hclust' in the 'stats' package [78]. We used the R package 'dynamicTreeCut' [79] to identify significant clusters, while specifying a minimum cluster size of 2 [80]. These methods for determining MHC supertypes have been shown to identify biologically relevant variation in MHC allele functionality in both human and non-human primate studies [69,81–84]. We calculated an individual's MHC supertype diversity as the number of unique MHC supertypes and calculated MHC supertype complementarity as the number of MHC supertypes shared between two individuals divided by the total number of unique supertypes possessed by the two individuals in total. Using our genome-wide metrics, as well as our allele-based and supertype-based MHC descriptors, we calculated in total 5 metrics of diversity and 5 metrics of genetic complementarity for each individual, summarized in Table I in S2 Appendix.

## Vaginal RNA sample collection

We collected vaginal RNA samples ($N = 307$ samples, $34.1 \pm 2$ samples per female) every other day throughout sexual skin tumescence and detumescence, and every 3 days throughout the rest of the cycle. To collect RNA samples, we inserted a sterile cotton swab into the vaginal opening (~2 to 3 inches) and rotated for 10 seconds. Once removed, we immediately placed the swab into a 1.5 mL DNA lo-bind collection tube containing 500 µl of Qiagen RNA Protect cell reagent and placed it into a cooler for transport back to the lab within one hour. When taking a post-copulatory sample, we removed any visible sperm plug from the vaginal opening using autoclaved forceps before inserting the cotton swab. We performed a piggyback centrifugation to transfer the vaginal cells in solution from the sample collection tube (containing the swab) into a new cryotube and froze at −80°C.

## Vaginal pH measurement

We measured the vaginal pH of each female daily ($N = 359$ measurements, $39.9 \pm 6.4$ measurements per female), using an ISFET probe and portable SI400 pH meter (Sentron). Prior to each sampling, we calibrated the probe using pH 4 and pH 7 buffers. The linear relationship between the raw voltage reading and the pH values of the known calibration solutions always fell between 95% and 105%, indicating proper function of the probe. We collected three sequential pH measures to determine an average pH reading for each female each day. In the case that a female was not cooperative in taking three separate readings, we instead took only two ($n = 7$) or one ($n = 5$). To do so, we inserted the probe into the vaginal opening (~2 to 3 inches) and waited for the reading to stabilize (~6 to 10 s) before recording the value. The ISFET probe simultaneously measures temperature and performs an automatic temperature compensation correction to account for differences in temperature between readings. We refrained from taking pH readings within 30 min following urination, and took post-copulatory measurements ($n_{post-cop} = 15$) immediately following post-copulatory RNA sampling, approximately 4 hours after an observed copulation with ejaculation. Between sequential readings from the same individual, we cleaned the probe using deionized water. Between individuals, we cleaned the probe with 70% ethanol and deionized water. The probe was stored overnight in a pH 7 buffer, as per the manufacturer's instructions.

## Vaginal cytology

We predicted the timing of ovulation using vaginal cytology. We collected vaginal swabs for cytological slides by inserting a sterile cotton swab into the posterior vagina and rotating it for 10 s before removal. We prepared slides by rolling the swab across a glass microscope slide, applying a spray fixative (CytoRAL), and staining slides with a commercially available simplified Harris-Schorr staining kit (Diagnoestrus; RAL Diagnostics). Vaginal epithelial cells undergo characteristic cyclical changes throughout the ovarian cycle, allowing cycle phase to be determined by the proportion of cell types present on each slide (Fig F in S1 Appendix) [85,86]. Approaching ovulation, white blood cells (WBCs) and mucus are present, and the proportion of large, geometric superficial cells gradually increases. Ovulation is detected by a sharp drop in the proportion of red-staining superficial cells, quantified by assessing the stained color of 100 cells and calculating the eosinophilic index (EI) as number of red cells + the number of red/blue (polychromatophilic) cells * 0.5 (Fig G in S1

Appendix) [87]. In addition to this quantitative measure, ovulation is also qualitatively associated with the disappearance of WBCs and mucus. The postovulatory phase is characterized by the return of WBCs and mucus, cellular clumping, and a return of basal and intermediate cell types.

From these patterns, we identified a 2-day ovulation window as the day of ovulation and the previous day. We then defined a 5-day fertile phase as the two days prior to and one day following the 2-day ovulation window [40]. We classified the 5 days preceding the fertile phase as the pre-fertile phase and the 5 days following the fertile phase as the post-fertile phase. This method for pre-fertile, fertile, and post-fertile phase classification is well established in the primatological literature, and has been used in numerous studies with respect to non-human primate sexual swellings and behavior [38–40]. The evaluation of cytological slides to determine ovarian cycle phase has been used with great success in this study population [88,89].

## Semen sample collection

To account for male-derived RNA present in post-copulatory vaginal RNA samples, we collected two masturbatory semen samples from each male (*N* = 8 samples) and conducted RNA sequencing. To do so, we collected coagulated semen left on the enclosure substrate immediately following an observed masturbation. We used autoclaved tweezers to place the sample into a 5mL lo-bind Eppendorf tube and immediately transported it back to the lab. Under a sterile fume hood, we removed the solid portion of the ejaculate, measured the volume of the remaining liquid portion using a pipette, added RNAprotect Cell Reagent in a volume 5 times the liquid sample volume, and froze at −80°C. All samples were frozen within 20 min following the time of ejaculation. We used semen samples collected after masturbation to minimize potential contamination from female-derived RNA. Furthermore, collection of semen from the female vaginal tract post-mating would have required disruption of the sperm plug, which was incompatible with later vaginal RNA sampling 4 hours later. Although the composition of ejaculates produced via masturbation may differ from those produced in a mating context, this approach represented the most feasible and controlled option for characterizing male-derived RNA in ejaculates.

## RNA extraction and sequencing

We extracted RNA from vaginal and semen samples using the Qiagen RNeasy Mini kit, according to the manufacturers' recommended protocol. We incorporated a preliminary PBS wash of the cells, used a QIAshredder for sample homogenization, performed an on-column DNase digestion to improve quality and concentrations, and measured RNA concentration and integrity using an Agilent TapeStation. Across all collected samples, vaginal sample concentrations ranged from 0.07 to 499.5 ng/μl (mean = 15.4 ± 2.7 ng/μl) and semen sample concentrations ranged from 0.005 to 0.5 ng/μl (mean = 0.18 ± 0.03 ng/μl). Library preparation and sequencing was performed at the University of Calgary's Centre for Health Genomics and Information sequencing core. We sequenced 106 vaginal samples and 8 semen samples with RIN values varying from 1.6 to 9.2 (mean = 5.3 ± 1.7 s.d.). We performed strand-specific library preparation using the NEBNext Ultra II RNA kit with rRNA depletion following the manufacturer's instructions and performed whole transcriptome sequencing on one NovaSeq6000 S2 100 cycle v1.5 run, generating 50 bp PE reads.

## Data processing of RNA-seq libraries

RNA sequencing generated an average of 38.1M (±21.4 s.d.) reads per sample. We trimmed and filtered reads for quality using the program Trimmomatic [90], with the following parameters: -phred33 LEADING:3 TRAILING:3 SLIDINGWINDOW:4:15 MINLEN:36. We used the splice-aware alignment tool STAR [91] to align sequences to the olive baboon reference genome (NCBI: GCA_008728515.1). Due to the degraded nature of our samples, we used the following parameters to allow for shorter alignments, as has been done with success in other studies: --outFilterScoreMinOverLread 0.3 --outFilterMatchNminOverLread 0.3 –outFilterMatchNmin 0 [92–94]. Due to issues with paired-end sequence alignment, substantially more reads displayed unique mapping in single-end versus paired-end mapping mode (paired-end mode:

3.8±0.56 million uniquely mapped reads per sample, single-end mode: 9.9±0.52 million uniquely mapped reads per sample). To maximize read counts mapped to genomic features, we used only R1 data for all analyses presented here. Studies have demonstrated an approximately 5% false positive and 5% false negative discovery rates for DE genes using single-end as opposed to paired-end reads [95]. These small discrepancies can exacerbate differences in identified gene ontology terms, with overlap between single- and paired-end data falling into the range of 40% [95]. To mitigate potential mapping errors due to the large bacterial cell populations present in the vagina, we additionally filtered mapped reads based on their taxonomic classification using Kraken 2 [96]. We built a custom database containing the olive baboon reference genome, as well as all bacterial, archaeal, fungal, protozoal, and viral genomes available through NCBI. We classified sequences using default kraken parameters, and filtered the STAR mapping results to include only reads that were either confidently classified as olive baboon or not classified as any other type of microorganism. Due to a high number of reads mapping to bacterial, archaeal, fungal, protozoal, and viral genomes, a mean of 27.8% of reads per sample (9.9M±5.6 s.d.) passed our kraken classification filter and were mapped uniquely to the baboon genome. From the filtered alignment files, we generated read counts of genomic features using the program Rsubread [97] and the Panubis1.0 genome annotation release 104.

## Modeling vaginal physiology across cycle phases

To examine how vaginal gene expression differs across cycle phases, we performed differential expression analysis using the edgeR package in R [98]. To do so, we first subset samples to include the 8 from each cycle phase (pre-fertile, fertile, post-fertile, and non-fertile) that had the highest number of uniquely mapped reads ($N=32$ samples total). Eight of the 9 study females are represented in the final set of 32 samples, with a mean of 6.25 females included within each cycle phase. By analyzing only a subset of all sequenced samples, we ensured an equal sample number across cycle phases and increased the mean number of uniquely mapped reads across samples from 9.9M to 13.7M. We filtered the list of genes included in the analysis by removing ribosomal protein genes, genes without human orthologs, and genes in which more than half of the samples had less than 10 counts per million, resulting in a mean library size of 4.9M read counts across 3,154 analyzable genes. We normalized library sizes based on the filtered gene list using the edgeR function 'calcNormFactors' and transformed count data for linear modeling using the 'voomWithQualityWeights' function in the package 'limma' [42]. To model our data, we controlled for female ID using the limma function 'duplicateCorrelation' with female ID as a blocking variable and included sample RIN and male presence (whether or not the sample was taken during the month prior to male introduction or after the male had been introduced) as covariates. We fit a linear model for each gene using the function 'lmFit' and stabilized the variance estimates across genes by applying a robust empirical Bayes moderation to the standard errors of the fitted coefficients using the function 'eBayes' and argument 'robust=TRUE'. We then applied empirical Bayes adaptive shrinkage using the 'ash' function from the 'ashr' package in R, which borrows information across genes to shrink effect size and uncertainty estimates towards zero, generate more robust posterior estimates, and calculate local false sign rate (LFSR) [43]. LFSR is a measure which integrates both effect size and certainty to generate a posterior probability that an estimated effect is in the correct direction (positive or negative). This approach is particularly advantageous in small-sample settings when variance estimates are unstable because it downweights imprecise measurements and provides a more reliable alternative to standard FDR corrections [99]. We identified DE genes as those falling below a 10% LFSR, a cut-off which is standard in the field of genomics [100–102], and also report the number of DE genes at more stringent 5% and 1% cutoffs. We performed GSEA using the 'fgsea' function in the R package 'clusterProfiler' [103] and the *Papio anubis* Ensembl genome annotations available through biomaRt [104], using a *p*-value cutoff of 0.05.

To examine how vaginal pH changes across cycle phases, we conducted robust linear mixed modeling using the R package 'robustlmm' [44]. We included cycle phase as a categorical predictor variable with the non-fertile phase as the reference category, vaginal pH (averaged across the 3 measurements for that day) as the response variable, vaginal

temperature (averaged across the 3 measurements for that day) as a covariate, and female ID as a random effect. For this analysis, we included only pH measurements in which there was no observed mating or obvious signs of previous mating (*i.e.,* sperm plug present) that day (*N* = 275 measurements, 30.6 ± 5.41 s.d. per female, 68.8 ± 21.8 s.d. per cycle phase). Visual inspection of quantile–quantile and residual variance plots confirmed homoscedastic residual variance structure and variance inflation factor (VIF) <2 confirmed no issues of collinearity.

## Modeling vaginal physiology in response to mating

To examine how vaginal gene expression changes in response to mating, we again performed differential expression analysis using the edgeR package in R. We analyzed 25 post-copulatory and 30 non-copulatory RNA samples, all of which were taken from the pre-fertile, fertile, or post-fertile phases and had greater than 5M uniquely mapped reads. We were able to collect post-copulatory samples from six of the nine females, and thus limited our non-copulatory samples to those six females as well. This resulted in a mean of 4.3 post-copulatory samples per female and 5 non-copulatory samples per female (Table A in S2 Appendix), with 12.3M (± 7 s.d.) uniquely mapped reads for post-copulatory samples and 8.5M (± 4 s.d.) for non-copulatory samples. To account for male-derived RNA present in the vagina, we first removed genes found to be highly expressed in semen samples. From these samples, we identified 1,442 genes highly expressed in semen (average expression of >20 cpm, Table E in S2 Appendix), and removed these genes from all subsequent post-copulatory gene expression analyses. We then filtered the remaining genes by removing ribosomal protein genes, genes without human orthologs, and genes in which more than half of the samples had less than 10 counts per million, resulting in a mean library size of 1.9M read counts across 2,716 analyzable genes. We normalized library sizes based on the filtered gene list, controlled for dyad ID using a blocking variable, and fit a linear model applying a robust empirical Bayes moderation, including sample RIN, cycle phase, and male presence as covariates. The limma function 'duplicate-Correlation' supports the inclusion of only a single blocking factor, and thus we included dyad ID as the blocking variable as this uniquely identifies each male-female pair, capturing the repeated measures associated with both individual IDs. We identified DE genes as those falling below a 10% LFSR and performed GSEA as described above.

To examine how vaginal pH differs following mating, we conducted robust linear mixed effects modeling. We used a binary predictor variable (yes or no) indicating whether the pH measurement for that day was a post-copulatory or non-copulatory measurement (*N* = 62; $n_{post-cop}$ = 15, $n_{non-cop}$ = 47). Post-copulatory pH measurements were obtained from five out of nine females during their pre-fertile or fertile phase, thus we limited non-copulatory measurements to those same females and phases (Table A in S2 Appendix). We used pH as the response variable, phase and temperature as covariates, and dyad ID as a random effect. Visual inspection of a quantile-quantile plot revealed greater residual variance in post-copulatory versus non-copulatory samples, which we statistically confirmed with a Breusch-Pagan test using the 'bptest' function in the 'lmtest' package in R [105]. We used VIF to confirm no issues of collinearity (VIF < 2).

## Modeling vaginal physiology in relation to genetic diversity and complementarity

To test how post-copulatory gene expression is related to male genetic diversity and complementarity, we used the same subset of RNA-seq samples described above for our post-copulatory analyses. We fit 10 separate models, each testing for the interactive effect between post-copulatory status (yes or no) and one genotype metric (listed in Table I in S2 Appendix). We controlled for dyad ID using a blocking variable and ran a linear model with robust empirical Bayes moderation on the transformed counts including sample RIN, cycle phase, and male presence as covariates. We identified DE genes as those falling below a 10% LFSR and performed GSEA as described above. Because we tested genotype × post-copulatory status effects across 10 separate models, we applied an additional FWER correction using the Holm method to adjust the p-values for each gene across all models [106]. In the Results, we report how many genes with LFSR < 10% also meet a FWER-adjusted *p* < 0.05 after this correction.

To test whether post-copulatory vaginal pH is related to male genetic diversity and complementarity, we conducted robust linear mixed effects modeling using the same pH measurements as described above for our post-copulatory analyses. We ran 10 separate models, each testing for the interactive effect between post-copulatory status (yes or no) and one genotype metric (listed in Table I in S2 Appendix). We included cycle phase and temperature as covariates and dyad ID as a random effect. We confirmed homoscedastic residual variance structure by visually inspecting quantile–quantile and residual variance plots, and adjusted p-values for multiple hypothesis testing using the Holm FWER correction [106].

To evaluate the robustness of our findings to the influence of individual mating pairs, we conducted a leave-one-out sensitivity analysis. In this approach, we iteratively removed a single mating dyad from our dataset and reran our analyses testing for the effect of male genotype in modulating post-copulatory vaginal gene expression and pH. For each iteration, we recorded the number of DE genes and the interaction effect size estimate and associated confidence intervals. This method allowed us to identify potentially influential observations and quantify the overall stability of our results.

## Supporting information

**S1 Appendix. Fig A. GSEA pathways enriched for differential expression in the fertile phase.** The "activated" panel represents pathways enriched for genes with heightened expression in the fertile compared to non-fertile phase and the "suppressed" panel represents pathways enriched for genes with lower expression in the fertile compared to non-fertile phase. Dot size represents the number of genes represented within that pathway and the darker colors represent lower p-values. The data underlying this figure are provided in S5 Data. **Fig B. Vaginal pH did not vary significantly between cycle phases.** Error bars represent model predictions ± one standard error and points represent the raw data. The data underlying this figure are provided in S5 Data. **Fig C. GSEA pathways enriched for differential expression post-copulation.** Three gene set pathways are enriched for genes with heightened expression in post-copulatory versus non-copulatory contexts. Dot size represents the number of genes represented within that pathway and the darker colors represent lower p-values. The data underlying this figure are provided in S5 Data. **Fig D. GSEA pathways enriched for differential expression in relation to MHC I allelic complementarity.** Pathways enriched for genes with heightened expression after mating with males with low complementarity. Dot size represents the number of genes represented within that pathway and the darker colors represent lower p-values. The data underlying this figure are provided in S5 Data. **Fig E. Leave-one-out sensitivity analysis of vaginal pH model estimates.** Shown are estimated interaction effects between post-copulatory status and male genotype on vaginal pH across leave-one-out iterations. Points represent the estimated interaction term (post-copulatory status × male genotype), and error bars denote 95% confidence intervals. The genotype included in each model is indicated in the facet title. The data underlying this figure are provided in S5 Data. **Fig F. Vaginal epithelial cells stained using a modified Harris-Schorr technique.** Epithelial cell types include basal cells (A), intermediate cells (B), polychromatophilic superficial cells (C), and eosinophilic superficial cells (D). The preovulatory phase (E) exhibits a gradual increase in the proportion of red to blue staining cells, and the post-ovulatory phase (F) is characterized by an increase in cellular clumping, mucus, and WBCs. **Fig G. Composite profile demonstrating fluctuations in EI over the course of the ovarian cycle ($N=22$ cycles).** Points represent mean EI values on each day in relation to ovulation, and error bars represent the standard error of the mean. The two-day ovulation window is designated by consecutive 0's on the x axis and is shaded in red. The days leading up to ovulation are designated by negative numbers and the days following ovulation designated by positive numbers. The data underlying this figure are provided in S5 Data. (DOCX)

**S2 Appendix. Table A. Description of dataset composition for vaginal pH and gene expression analyses.** Total number of samples, as well as number of samples per cycle phase or copulatory status for cycle phase and post-copulatory analyses, respectively. Measures of male genetic heterozygosity and complementarity are described in Table 1 in S2 Appendix. **Table B. Differentially expressed genes- between phases. Genes with significant (FDR<10%)**

differential expression in pairwise comparisons between cycle phases. Negative coefficients correspond to lower expression in the phase listed first in the comparison column, and positive coefficients correspond to higher expression in the phase listed first. **Table C. Gene set enrichment analysis- between phases.** GO pathways that are overrepresented among genes that are differentially expressed between the fertile and non-fertile phase. **Table D. Vaginal pH linear model results- cycle phase. Table E. Genes highly expressed in semen.** Genes with an average of >20 cpm in semen samples. **Table F. Differentially expressed genes- post-copulatory versus non-copulatory.** Genes with significant (LFSR < 10%) differential expression in post-copulatory versus non-copulatory samples. Negative coefficients correspond to lower expression post-copulation and positive coefficients correspond to higher expression post-copulation. **Table G. Gene set enrichment analysis- post-copulatory versus non-copulatory.** GO pathways that are overrepresented among genes that are differentially expressed between post-copulatory versus non-copulatory samples. **Table H. Vaginal pH linear model results- post-copulatory status. Table I. Measures of male genetic diversity and dyadic complementarity. Table J. Differentially expressed genes- male genetic diversity.** Genes with a significant (FDR < 10%) interactive effect between post-copulatory status and one metric of male genetic diversity (listed in the "genotype metric" column). **Table K. Differentially expressed genes- genetic complementarity.** Genes with a significant (FDR < 10%) interactive effect between post-copulatory status and one metric of genetic complementarity (listed in the "genotype metric" column). **Table L. Gene set enrichment analysis- male genetic diversity and complementarity.** Pathways with a significant enrichment (FDR < 10%) interactive effect between post-copulatory status and one metric of male genetic diversity or complementarity (listed in "genotype metric" column). **Table M. Number of genes/pathways whose post-copulatory expression is significantly modified by an aspect of male genetic makeup across five mating dyads.** Number of differentially expressed (DE) genes at passing an LFSR < 0.1, LFSR < 0.05, and LFSR < 0.01 threshold, number of LFSR < 0.1 genes which also pass a family-wise error rate (FWER)-adjusted $p < 0.05$ threshold, and number of gene set enrichment analysis (GSEA) pathways at a $p < 0.05$ threshold. **Table N. Vaginal pH linear model results- male genetic diversity and complementarity.** Interactive effect of post-copulatory status and each measure of male genetic diversity or complementarity in predicting vaginal pH. Each genetic metric was tested in a separate model. **Table O. Primer sequences used to amplify MHC A, B, DQA, and DRB loci.**
(XLSX)

**S1 Data. Data underlying main figure Fig 1B–1F.**
(XLSX)

**S2 Data. Data underlying main figure Fig 2B–2F.**
(XLSX)

**S3 Data. Data underlying main figure Fig 3A–3D.**
(XLSX)

**S4 Data. Data underlying main figure Fig 4A–4C.**
(XLSX)

**S5 Data. Data underlying Figs A, B, C, D, E and G in S1 Appendix.**
(XLSX)

## Acknowledgments

We would like to thank members of the Primate Hormones and Behavior lab at NYU, the Primate Genetics Lab at DPZ, and the Melin lab at the University of Calgary for their support in completing this work. We extend a huge thank you to all of the staff at the CNRS Station de Primatologie for their assistance in executing this project, specifically Romain Lacoste,

Slaveia Garbit, Magali Ghirart, Pascaline Boitelle, and Pau Molina. Thank you to Beth Archie and Cliff Jolly for their value feedback throughout the formulation and execution of this project. Thank you to Kristi Holt for collecting data and Stefano Vaglio for his insight into baboon training. Thank you to Patrícia Ströher, Gwen Duytschaever, and the University of Calgary's Centre for Health Genomics and Informatics sequencing core for facilitating RNA preparation and sequencing. This work was supported in part through the NYU IT High Performance Computing resources, services, and staff expertise.

## Author contributions

**Conceptualization:** Rachel M. Petersen, James P. Higham.

**Data curation:** Rachel M. Petersen.

**Formal analysis:** Rachel M. Petersen.

**Funding acquisition:** Rachel M. Petersen, James P. Higham.

**Investigation:** Rachel M. Petersen, Lee (Emily) M. Nonnamaker, Jaclyn A. Anderson.

**Methodology:** Rachel M. Petersen, Christina M. Bergey, Christian Roos, Amanda D. Melin, James P. Higham.

**Visualization:** Rachel M. Petersen.

**Writing – original draft:** Rachel M. Petersen.

**Writing – review & editing:** Rachel M. Petersen, Christian Roos, Amanda D. Melin, James P. Higham.

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
