## [Editor Report · Decision Letter 0]

6 Sep 2025

Dear Dr Petersen,

Thank you for submitting your manuscript entitled "Genetically-based sperm discrimination in the vaginal tract of a primate species" for consideration as a Research Article by PLOS Biology.

Your manuscript has now been evaluated by the PLOS Biology editorial staff as well as by an academic editor with relevant expertise and I am writing to let you know that we would like to send your submission out for external peer review.

Once your full submission is complete, your paper will undergo a series of checks in preparation for peer review. After your manuscript has passed the checks it will be sent out for review. To provide the metadata for your submission, please Login to Editorial Manager (https://www.editorialmanager.com/pbiology) within two working days, i.e. by Sep 09 2025 11:59PM.

Kind regards,

Ines

--

Ines Alvarez-Garcia, PhD

Senior Editor

PLOS Biology

---

## [Decision Letter · Decision Letter 1]

24 Oct 2025

Dear Dr Petersen,

Thank you for your patience while your manuscript entitled "Genetically-based sperm discrimination in the vaginal tract of a primate species" was peer-reviewed at PLOS Biology. The manuscript has now been evaluated by the PLOS Biology editors, an Academic Editor with relevant expertise, and by two independent reviewers.

The reviews are attached below. As you will see, while we are mostly satisfied with how you have addressed the previous issues raised by the associated reviews you shared with us, we were wondering about the effect the small sample size has in your results, as this was not covered by the previous reviewers. Thus, the two new reviewers looked only at this aspect and find the conclusions potentially interesting, but they also raise concerns about the sample size that should be addressed before we can consider the manuscript for publication. Reviewer 1 thinks that the high-dimensional RNAseq data is highly underpowered and might lead to an increased sensitivity to outliers. This reviewer also notes a lack of adjustment for the testing across models and that interactions normally require even a higher sample size to be able to compare them to main effects. In addition, the reviewer thinks you should provide effect size estimates with confidence intervals for all major findings. Reviewer 2 concludes that the sample is quite small to validate the findings, and that the word ‘sample’ is confusing throughout the manuscript and should be clarified. The reviewer suggests using N to denote the whole sample in all cases and toning down the claims, stating clearly the limitations of the sample size, which should be clearly discussed in the manuscript, including the abstract.

In light of the reviews, we would like to invite you to revise the work to thoroughly address the reviewers' reports. However, we would like to consider the manuscript as a Discovery Report, thus please select that format from the drop down when you submit the revision. Discovery Reports only allow 4 main figures, but since your manuscript has that exact number, the change of format will be only nominal and won't require any changes. Your revised manuscript is likely to be sent for further evaluation by all or a subset of the reviewers.

**IMPORTANT - SUBMITTING YOUR REVISION**

3. Resubmission Checklist

a) *PLOS Data Policy*

b) *Published Peer Review*

Sincerely,

Ines

--

Ines Alvarez-Garcia, PhD

Senior Editor

PLOS Biology

Reviewers' comments

Rev. 1:

The manuscript tackles an important and underexplored question. However, the statistical evidence is fragile due to small sample size, multiple testing, and reliance on interaction models. Results should be interpreted as preliminary, and stronger claims of genetically-based sperm discrimination in primates are not statistically justified at present.

Major Concerns

- The study sample size is very small producing limited dyads. For high-dimensional RNA-seq data, this looks like it is highly underpowered and may lead to increased sensitivity to outliers.

- Thousands of genes are tested assuming a10% FDR: how do you justify such a high level? Given small n, variance estimates are unstable, raising the risk of false discoveries.

- Several measures of genetic diversity/complementarity are evaluated. Although FDR is controlled within each analysis, no adjustment for the family-wise testing across all these models is provided.

- Mixed models with few levels for random effects and limited sample sizes are highly instable and reliable variance component estimation is questionable.

- Interaction terms between post-copulatory status and genotype are the central findings. Interactions notoriously require even higher sample size to be evaluated compared to main effects, and this study is clearly underpowered. The small sample size makes effect estimates hard to believe.

- The post-copulatory pH dataset is very limited (15 measurements). Linear mixed modeling with multiple covariates on this dataset is likely over-parameterized.

- Variance tests (Breusch-Pagan) with such small samples may not be robust to distributional assumptions.

- Provide effect size estimates with confidence intervals for all major findings (gene expression contrasts, pH changes, genotype interactions).

Rev. 2:

Overall, we think the statistical methods used in the manuscript are appropriate; the major issue is the sample sizes, or so-called 'large p small n problem'. No matter whether this manuscript will be published or not, we must thank the authors for their efforts and time on the experiments, which are impressive, expansive and exciting to analyze. In the following, not all comments need a response - some are purely for communicating statistical ideas or information; in that case, they are obvious.

In the 'Genome-wide and MHC genotyping section', we see that the sample sizes (N=13, n=4 males, and n=9 females) are small; when they are divided into subset groups, n is even smaller for each (refer to line 526 for different cycles); so we suggest that the authors would use very conservative languages to communicate their conclusions for the current study and remind the readers that all the conclusions are tentative therefore need larger sample experiments to validate.

The word sample is quite confusing throughout the manuscript. We should distinguish between the whole sample size (N as it usually is used in statistics, representing the population that was being studied) and the samples or sample sizes that were a subset of the whole, for example RNA samples (refer to the line 450). So, we'd suggest that the authors add a capital N to the whole sample such as N=13 or whatever is more appropriate in other cases (when N is number of animals, such as N1=9 or N2=4 etc. So that N1+N2+... =13).

Since N is small, the evidence from the analyses on the results wouldn't be very strong and the validity of the whole study could be greatly reduced, therefore not truly convincing our readers. In other words, if N is smaller than a dozen, then all the statistical models could be invalid, whether it is a mixed model or a simple linear model because there is virtually no degrees of freedom that was left for variation estimation in the model, which will include huge variations, for example, a very wide 95% CI (or large width) on the mean or median value of something measured. Intuitively and statistically approved, it's hard to believe anything deduced from a small sample. Genes, cycles etc. are secondary level variables (derived) in the current study, the first level ones (root, original) are the number of animals, the IDs, the male or female etc., as we generally see from a so-called demographic table (the 1st table usually) in a paper.

On lines 86 to 88, the authors mentioned the number of RNA samples and related number of measurements on those samples, but we'd love to see the number of animals, which is more significant in the current study. And besides, the sample size of the study should be reflected in the ABSTRACT section, where we did not see either. In addition, on line 60 and following lines, we did not observe a description on the sample size of the current study. On lines 84 to 88, we did not observe a description either. On line 176, we should have seen a sample size number, but we did not.

Refer to lines 89 and following lines, line 133 and following lines. Since the authors used ID as block factor and it will consume degrees of freedom of the data for each ID included, therefore reduced the effective sample size of the data. How was this issue considered in the processing of the data?

Line 104 and following lines, or line 141 and following lines - the authors used a linear mixed model. How were those variables selected? Any experimental (clinical) or statistical considerations

or a purely arbitrary choice?

On line 196, we see a very tiny p value, but this comes from a very small sample size of study, which is a very common dilemma in a genetic study. Statistically, a very small p value should come from a large sample. And a common misunderstanding about the p value is that the statistical null hypothesis is that there is no difference or difference is zero, a p value only tells us that the difference is different from zero (not zero), but how different is not solved. For example, the difference of 0.002 is different from zero, and 0.02 or 0.2 is different from zero but they are 10 or 100 times larger or smaller comparing one to another.

On line 211, the numbers look reasonable, but on line 214, we see a different scenario: the range is huge, because this is small sample size, and anything derived from a small sample would be an unstable sample, does not follow a normal distribution (when median is close to the mean value), which is why people have to rely upon a median value (like median income); however, a median is farther away from the truth (the mean is better, at least a trend, the characteristic or the majority of something). So, we need a large sample to validate a conclusion generally.

In Table 1 and in Figure 3, again, we don't know the sample sizes, so we are not convinced that the results are good to generalize to other cases or external data, this is, we are not certain whether we should accept the conclusions.

In Figure S5, we are not sure what that "p < 0.05" is, please add an explanation at the footnote.

---

## [Decision Letter · Decision Letter 2]

18 Feb 2026

Dear Dr Petersen,

Thank you for your patience while we considered your revised manuscript entitled "Evidence for genetically-based sperm discrimination in the vaginal tract of a primate species" for publication as a Discovery Report at PLOS Biology. This revised version of your manuscript has been evaluated by the PLOS Biology editors, the Academic Editor and one of the original reviewers.

Based on the review, we are likely to accept this manuscript for publication, provided you satisfactorily address the data and other policy-related requests stated below my signature.

In addition, we would like you to consider a suggestion to improve the title:

"A female primate can exert post-copulatory selection via genetically-based sperm discrimination in the vaginal tract"

We expect to receive your revised manuscript within two weeks.

*Published Peer Review History*

*Press*

Sincerely,

Ines

--

Ines Alvarez-Garcia, PhD

Senior Editor

PLOS Biology

DATA POLICY:

Fig. 1D-F; Fig. 2C-F; Fig. 3A-D; Fig. 4A-C; Fig. S1; Fig. S2; Fig. S3; Fig. S4; Fig. S5 and Fig. S7

CODE POLICY

Per journal policy, if you have generated any custom code during the course of this investigation, please make it available without restrictions. Please ensure that the code is sufficiently well documented and reusable, and that your Data Statement in the Editorial Manager submission system accurately describes where your code can be found. More information on our Code Policy, what and how to share can be found here: https://journals.plos.org/plosbiology/s/code-availability

SPECIES INDICATED IN THE ABSTRACT?

- Please note that per journal policy, the model system/species studied should be clearly stated in the abstract of your manuscript.

Reviewers' comments

Rev. 2:

We are happy to read the revised manuscript. Thank you very much for your efforts and time to revise the mansucript by following carefully the previous comments and answered each questions with new evidence and methods in some cases.

---

## [Editor Report · Decision Letter 3]

24 Feb 2026

Dear Dr Petersen,

Thank you for the submission of your revised Discovery Report entitled "Evidence for genetically-based sperm discrimination in the vaginal tract of a primate species" for publication in PLOS Biology. On behalf of my colleagues and the Academic Editor, Masahito Ikawa, I am delighted to let you know that we can in principle accept your manuscript for publication, provided you address any remaining formatting and reporting issues. These will be detailed in an email you should receive within 2-3 business days from our colleagues in the journal operations team; no action is required from you until then. Please note that we will not be able to formally accept your manuscript and schedule it for publication until you have completed any requested changes.

PRESS

Sincerely,

Ines

--

Ines Alvarez-Garcia, PhD

Senior Editor

PLOS Biology
